# Theta-burst microstimulation in the human entorhinal area improves memory specificity

Ali S Titiz[1†], Michael R H Hill[1,2†], Emily A Mankin[1†], Zahra M Aghajan[3], Dawn Eliashiv[4], Natalia Tchemodanov[1], Uri Maoz[1,2,5§], John Stern[4], Michelle E Tran[1], Peter Schuette[3], Eric Behnke[1], Nanthia A Suthana[1,3,5‡], Itzhak Fried[1,3‡*]

[1]Department of Neurosurgery, University of California, Los Angeles, United States; [2]California Institute of Technology, Pasadena, United States; [3]Department of Psychiatry and Biobehavioral Sciences, University of California, Los Angeles, United States; [4]Department of Neurology, University of California, Los Angeles, United States; [5]Department of Psychology, University of California, Los Angeles, United States

**\*For correspondence:**
ifried@mednet.ucla.edu

[†]These authors contributed equally to this work
[‡]These authors also contributed equally to this work

**Present address:** [§]Chapman University, Orange, United States

**Competing interests:** The authors declare that no competing interests exist.

**Abstract** The hippocampus is critical for episodic memory, and synaptic changes induced by long-term potentiation (LTP) are thought to underlie memory formation. In rodents, hippocampal LTP may be induced through electrical stimulation of the perforant path. To test whether similar techniques could improve episodic memory in humans, we implemented a microstimulation technique that allowed delivery of low-current electrical stimulation via 100 $\mu m$-diameter microelectrodes. As thirteen neurosurgical patients performed a person recognition task, microstimulation was applied in a theta-burst pattern, shown to optimally induce LTP. Microstimulation in the right entorhinal area during learning significantly improved subsequent memory specificity for novel portraits; participants were able both to recognize previously-viewed photos and reject similar lures. These results suggest that microstimulation with physiologic level currents—a radical departure from commonly used deep brain stimulation protocols—is sufficient to modulate human behavior and provides an avenue for refined interrogation of the circuits involved in human memory.

DOI: https://doi.org/10.7554/eLife.29515.001

## Introduction

The hippocampus has been implicated in episodic and autobiographical memory formation in animal models (*Devito and Eichenbaum, 2011*; *Ergorul and Eichenbaum, 2004*; *Morris et al., 1982*; *Squire, 1992*) and humans (*Squire and Zola-Morgan, 1991*; *Tulving, 2002*). Using electrical stimulation to modulate hippocampal activity to restore memory function has recently become a focus of considerable interest. This idea is motivated to an extent by a long history of research on long-term potentiation (LTP), which is accepted as a neural substrate of learning and memory (*Bliss and Lomo, 1973*; *Kandel and Schwartz, 1982*) and can be induced by electrical stimulation to hippocampal afferents, such as the perforant path or Schaffer collaterals (*Bliss and Lomo, 1973*; *Cooke and Bliss, 2006*; *Larson et al., 1986*; *Nguyen and Kandel, 1997*; *Staubli and Lynch, 1987*).

Deep brain stimulation (DBS) of hippocampal-related targets, such as the fornix (*Hamani et al., 2008*; *Miller et al., 2015*) and the entorhinal region (*Suthana et al., 2012*), have shown promise for modulating human cognition. However, DBS is commonly applied through pairs of large clinical electrodes (e.g., 1.27 mm diameter), which lack target specificity and, hence, may affect wide and

heterogeneous neuronal assemblies in nearby brain regions. More spatially-focused stimulation in the medial temporal lobe (MTL) could be achieved using smaller electrodes (for example, 100 μm diameter microelectrodes), allowing higher anatomical specificity and physiological modulation of brain circuits critical for learning and memory. Moreover, targeting afferent projections may be a more specific and efficient strategy to drive downstream neurons (*Riva-Posse et al., 2014*), as suggested by recent optogenetic studies (*Gradinaru et al., 2009*; *Rajasethupathy et al., 2016*).

Microstimulation mimics effects that drive neurons naturally (*Fetsch et al., 2014*) and has been shown in animal studies to modulate a variety of specific brain functions, depending on the site of stimulation (*Hamani et al., 2008*; *Histed et al., 2009*; *Logothetis et al., 2010*). For example, in non-human primates, stimulation using microelectrodes improved face categorization (*Afraz et al., 2006*), modulated motion perception with great specificity (*Fetsch et al., 2014*), and increased the learning rate during a reinforcement learning task (*Williams and Eskandar, 2006*). In humans, microstimulation of primary visual cortex has been shown to induce phosphenes (*Schmidt et al., 1996*), and microstimulation of the substantia nigra can influence reinforcement learning (*Ramayya et al., 2014*). Further evidence of the high specificity that this method of stimulation can bring to human brain modulation has been provided by intraoperative investigations (*Histed et al., 2013*; *Logothetis et al., 2010*). Stimulation using microwires presents a novel opportunity to improve selective and natural activation of neuronal circuits in the human brain.

Here, we asked whether application of microstimulation targeted to the entorhinal afferents into hippocampus could enhance declarative memory function in humans. We postulated that afferent stimulation would most directly affect the downstream hippocampal fields, and thus memory function as well. Importantly, LTP induction by high-frequency microstimulation of the perforant path in vivo has been shown to cause reorganization of wide hippocampal and cortical networks in rats (*Canals et al., 2009*). A theta-burst protocol has been shown to be optimal for inducing LTP (*Larson et al., 1986*). We therefore hypothesized that theta-burst microstimulation, targeted to brain regions containing afferent fibers to the hippocampus, would improve episodic memory performance in humans.

One crucial aspect of episodic memory is pattern separation, the ability to retrieve the specifics of past events without generalizing to similar or partially overlapping events (*Yassa and Stark, 2011*). During a task in which subjects studied photos of novel people, microstimulation was applied in a theta-burst pattern. We show that theta-burst microstimulation of the right entorhinal area, applied prior to stimulus onset, enhanced memory specificity for these photographs. That is, during a test phase, subjects performed better at accepting photographs from the encoding stage as previously viewed, concomitant with rejecting similar (lure) photographs.

## Results

Participants were thirteen neurosurgical patients with pharmacoresistant epilepsy, implanted with intracranial depth electrodes, who performed a person recognition task (*Figure 1*, *Figure 1—source data 1*). During the encoding phase, subjects viewed novel portraits of people. On half of the trials, randomly selected for each participant, electrical stimulation was applied during the fixation period prior to stimulus onset (see *Microstimulation Protocol* section in Materials and methods) (*Figure 1A–B*). Stimulation was delivered through a 100 μm diameter microwire that had been targeted to the left or right entorhinal area (two subjects received stimulation on each side (in separate sessions); *Figure 1—source data 1*, *Figure 2*). In *post hoc* analysis, all stimulation microwires were determined to have been located in regions known to include afferent inputs to the hippocampus, including entorhinal white matter (angular bundle), entorhinal gray matter, or subiculum (*Figure 2*, *Figure 2—figure supplements 1–2*, *Figure 1—source data 1*) (*Yassa et al., 2010*; *Zeineh et al., 2017*). Because the subiculum contains—in fact, is perforated by—fibers of the perforant path, a major output of the entorhinal cortex, it was included in our sample. After the encoding phase, subjects performed a 30 s distractor task and then were presented with a series of images and asked whether each was 'old' (presented during the encoding phase) or 'new.' For each image presented during the encoding phase, two images were included in the test phase; one, the 'target' was the exact same image; the other, the 'lure,' was a portrait of a different person who looked similar to the person in the target portrait (*Figure 1A,C*). These images were presented one at a time, in pseudo-random order, during the test phase. Whenever possible, subjects performed the task more than once,

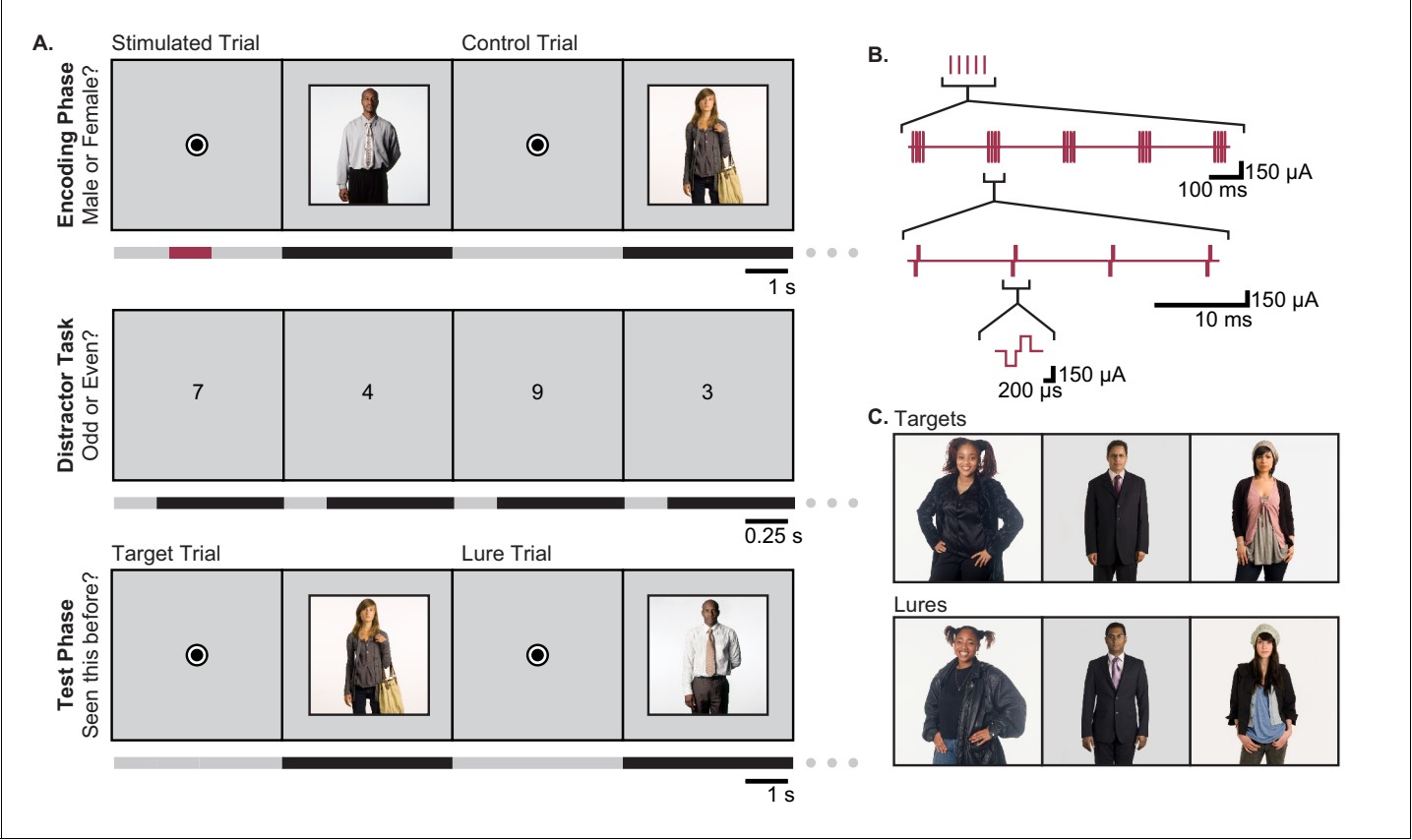

**Figure 1.** Task Design. (A) The person-recognition task consisted of three phases. In the encoding phase, participants were shown portraits of people for 4 s each (black bars in timeline), preceded by a 4.2–5.2 s fixation dot (gray bars). Half of the pictures were randomly selected for stimulation. For these, one second of theta-burst stimulation (red bar) was applied beginning 2.2–2.7 s before picture onset, during viewing of the fixation dot. To ensure that participants were viewing the portraits, they were asked to report whether the person in each image was male or female. The encoding phase was followed by a 30 s distractor task in which single digits were presented once per second and the participant was asked to identify each digit as odd or even. Finally, during the test phase, participants were shown a mixture of images they had seen during the encoding phase (targets) and portraits of people who looked similar but had not been previously viewed (lures). Participants were asked to report whether each image was 'old' or 'new.' (B) Theta burst stimulation consisted of 5 sets of current pulses, separated by 200 ms, in which each set included four biphasic stimulation pulses presented at 100 Hz. (C) Representative pairings of target and lure images. All images used in the task were adapted with permission from the book *Exactitudes* (**Versluis and Uyttenbroek, 2002**). Further examples of images may be viewed at http://exactitudes.com. Demographic data for the study participants are presented in *Figure 1—source data 1*.

DOI: https://doi.org/10.7554/eLife.29515.002

The following source data is available for figure 1:

**Source data 1.** Participant demographics and stimulation location.

DOI: https://doi.org/10.7554/eLife.29515.003

using a new set of images each time, such that a total of 40 experimental sessions were conducted (*Figure 1—source data 1*; Materials and methods).

To determine the behavioral effects of microstimulation, we defined several performance metrics. For each image presented during encoding, we noted whether it was recognized as 'old' during the test phase, as well as whether its corresponding lure image was correctly identified as 'new.' For each session, we computed the target acceptance rate as the proportion of target images that were correctly identified as old, and the lure rejection rate as the proportion of lure images that were correctly identified as new. Independently, these measures are not fully informative; for example, if a subject has a bias toward answering 'old', that would inflate the target acceptance rate while decreasing the lure rejection rate. Thus, a comparison of responses to targets and lures is required. On the whole-session level, we can compute the discrimination index (DI), which is the probability of answering 'old' on target images minus the probability of answering 'old' on lure images. DI works

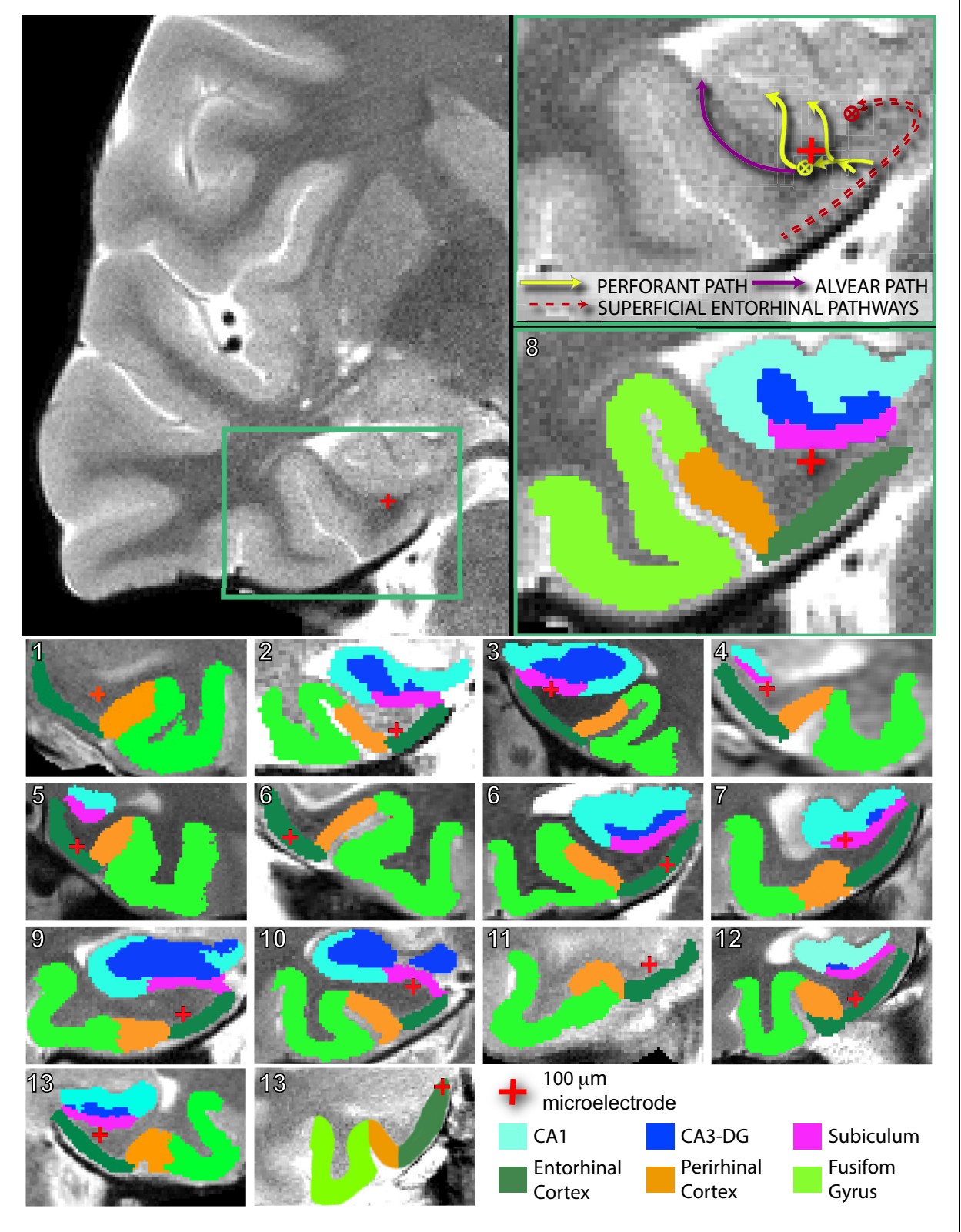

**Figure 2.** Electrode localization in participants. (Top) Pre-implantation high-resolution MRI scans (left) were co-registered to post-implantation high-resolution CT scans (not pictured) for electrode localization. Automated protocols were used to delineate different regions (colored areas in bottom right magnification), as well as white versus gray matter areas. The red crosshair denotes the location of the microstimulation electrode in the right angular bundle. Arrow diagrams, shown in the upper right magnification, depict the expected trajectories and directions of afferent pathways to the

*Figure 2 continued on next page*

*Figure 2 continued*

hippocampus; adapted from *Zeineh et al. (2017)*. ⊗ denotes fiber tracts traveling transverse to the coronal plane. Perforant path fibers begin in entorhinal gray matter, travel through the angular bundle, and pass through the subicular pyramidal layer, prior to arriving to either the dentate gyrus or CA fields of the hippocampus. (Bottom) Segmented MRIs and locations of microelectrodes, same as above, for all other participants in the study. Numbers in the upper left refer to participant ID (*Figure 1—source data 1*). Subregions were delineated by hand for participant 13's right hemisphere, as the electrode was located farther anterior than the automated protocols are intended to compute. A comparison of the spatial specificity of microstimulation and macrostimulation is shown in *Figure 2—figure supplement 1*. Group level localization data are presented in *Figure 2—figure supplement 2*.

DOI: https://doi.org/10.7554/eLife.29515.004

The following figure supplements are available for figure 2:

**Figure supplement 1.** Microstimulation electrodes provide spatially focused stimulation.

DOI: https://doi.org/10.7554/eLife.29515.005

**Figure supplement 2.** Group-level microelectrode placements.

DOI: https://doi.org/10.7554/eLife.29515.006

to subtract out the bias toward one answer or another, but does not explicitly compare performance for specific targets and their corresponding lures. Thus, for each image we assigned a behavioral label of 'remembered' or 'missed'. 'Remembered' images were those that were accepted as previously viewed, along with rejection of the corresponding lure. This required not only recognizing the target image but also remembering the details sufficiently to distinguish it from the lure image. We computed the remembered rate as the proportion of target images that met these criteria. All other images were categorized as 'missed.'

To evaluate how microstimulation affected these behavioral metrics, we computed each metric separately for the subset of stimulated trials and the subset of non-stimulated trials within each session and used generalized estimating equations (GEEs) to model the effects of stimulation condition on these measures. GEEs are a class of generalized linear models that were developed for analyzing repeated measures data (in our case, multiple experiments for an individual subject) and, contrary to the conventional repeated measures ANOVA tests, can handle different numbers of observations per subject and do not assume equal correlations between within-subject observations (*Gardiner et al., 2009*; *Gueorguieva and Krystal, 2004*; *Hubbard et al., 2010*; *Subramanian and O'Malley, 2010*), thus making this a rigorous statistical approach for our study (see Materials and methods for details)

We hypothesized that stimulation would improve memory specificity, but the effects of stimulation might vary with the precise location of the stimulating electrode. In particular, because previous studies in both human and non-human primates have indicated that facial processing may be lateralized within the hippocampus (*Fried et al., 1997*; *Haxby et al., 1996*), we predicted that stimulation within the left or right entorhinal area may be differentially effective. Thus, using GEEs, we modeled the difference in proportion of remembered pictures between stimulated and non-stimulated trials as a function of stimulation hemisphere. We found that there was a significant effect of stimulation hemisphere on the degree to which stimulation changed the proportion of remembered portraits (p=$6.17 \times 10^{-4}$, Wald $\chi^2$ = 11.72). Specifically, stimulation of the right entorhinal area significantly improved performance (Estimated Mean (EM) = 0.12, 95% CI [0.05, 0.18]), while stimulation of the left entorhinal area had no effect (EM = −0.017, 95% CI [−0.060, 0.025]) (*Figure 3A*, *Figure 3—source data 1*, *Source code 1*). We confirmed that these results were robust to the precise statistical method chosen; even a basic t-test on the difference between mean performance on stimulated and nonstimulated trials per participant revealed an effect of stimulation hemisphere (t(13) = −2.98, p=0.011), and *post hoc* t-tests indicated that right entorhinal stimulation caused significant improvement (mean change = 0.12 ± 0.04, t(8) = 3.53, p=0.008; effect size: Cohen's $d_z$ = 1.18, CL = 0.88), while left entorhinal stimulation did not (mean change = −0.02 ± 0.03, t(5) = −0.77, p=0.48) (*Figure 3B*). Further, out of 9 subjects who received microstimulation in the right entorhinal area, eight had a higher remembered rate for stimulated trials compared to nonstimulated trials, which is more than expected by chance (p=0.020, binomial test). Finally, because our subjects were patients with epilepsy, we wanted to ensure that the effect of stimulation was not driven by epileptic activity. To test this, we applied the GEE model described above to the subset of data that included only sessions in which the stimulating electrode was located outside of the seizure onset zone. Our results

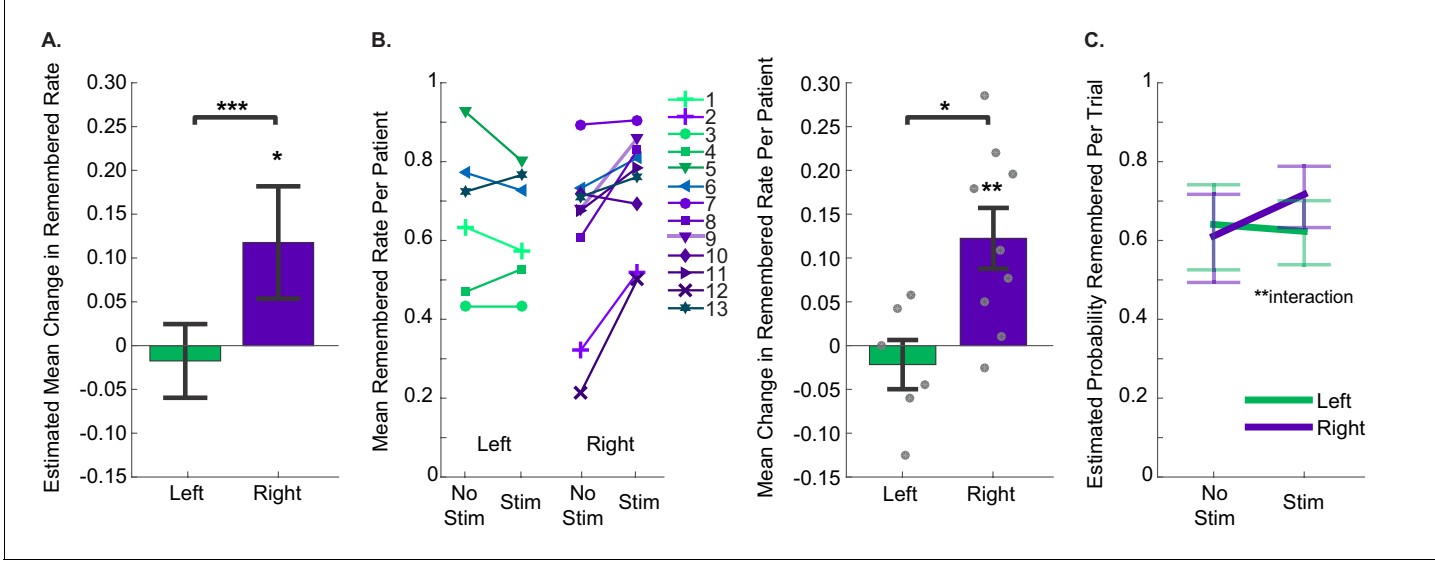

**Figure 3.** Stimulation of the right entorhinal area improved memory specificity. (**A**) The estimated mean change in fraction of stimulated items that were remembered compared to non-stimulated items, as estimated with Generalized Estimating Equations, which accounted for within subject correlations in performance. Positive numbers indicate that stimulated trials were remembered at greater rates. The effect of stimulation hemisphere was significant (p=6.17 × $10^{-4}$, N = 40 sessions from 13 participants (left: 21 sessions from 6 participants; right: 19 sessions from 9 participants)). Error bars indicate 95% confidence intervals, demonstrating that the increase in performance due to stimulation of the right entorhinal area was greater than 0. (Data and SPSS model definitions: *Figure 3—source data 1*, *Source code 1*.) (**B**) (Left) Remembered rates for each individual participant. For participants who did the task more than once, average rates are presented. Participants 6 and 13 performed the task with stimulation on each side (in different sessions), and the sessions from each hemisphere are presented independently. Upward slanting lines correspond to positive change scores, indicating that stimulation improved performance. (Right) Difference in remembered rates for left and right entorhinal area stimulation (mean ±s.e.m.). These are significantly different (t(13) = −2.98, p=0.011), and stimulation in the right entorhinal area leads to significantly positive changes in remembering (t(8) = 3.53, p=0.008). (**C**) Trial by trial analysis of whether a portrait was subsequently remembered was evaluated with a GEE model including stimulation condition and stimulation hemisphere, as well as trial-order effects. The only significant predictor was the interaction term between stimulation condition and hemisphere (p=0.002, N = 1207 trials from 13 participants). Error bars are 95% Wald Confidence Intervals. (Model coefficients: *Figure 3—source data 3*, data: *Figure 3—source data 2*, and SPSS model definitions: *Source code 2*).

DOI: https://doi.org/10.7554/eLife.29515.007

The following source data is available for figure 3:

**Source data 1.** Session-level performance data.
DOI: https://doi.org/10.7554/eLife.29515.008
**Source data 2.** Trial-level performance data.
DOI: https://doi.org/10.7554/eLife.29515.009
**Source data 3.** Trial by trial analysis of behavioral data.
DOI: https://doi.org/10.7554/eLife.29515.010

remained qualitatively the same, with a strong effect of stimulation hemisphere on behavior, and significant improvement induced by right-sided stimulation (p=9.49 × $10^{-3}$, Wald $\chi^2$ = 6.73, right EM = 0.11; 95% CI [.029. 184]).

Although our experimental design included selecting at random which trials in each session received stimulation, we wanted to ensure that our results were independent of any potential trial order effects. We thus modeled the behavioral data on a trial by trial basis while accounting for the trial order as well as whether the target image was presented before the lure image during the test phase (1207 trials). Here, the model included main effects of normalized trial number, whether the target was presented first, stimulation condition (trial received stimulation or not), stimulation hemisphere (left or right), and an interaction term between the last two. We found that the interaction term between the stimulation condition and hemisphere had a strong significant effect on behavior (p=2.19 × $10^{-3}$, Wald $\chi^2$= 9.38), while neither of the order variables were a significant predictor (normalized trial number: p=0.19, presentation of target before lure: p=0.053). Consistent with the previous model, stimulation of the right entorhinal area increased the probability of remembering a

portrait to a greater degree than stimulation of the left entorhinal area (*Figure 3C*; *Figure 3—source data 2–3*, *Source code 2*).

Because the order of presentation of target and lure trended strongly toward a behavioral effect—memory was generally better when the target was presented before the lure—we wanted to ensure that the interaction between stimulation condition and hemisphere was independent of whether the target or lure had been presented first. We thus divided our dataset into trials in which the target was presented first (618 trials) and trials in which the lure was presented first (589 trials), and modeled behavioral outcome with normalized trial number, stimulation condition, stimulation hemisphere and an interaction for each dataset. Consistent with the full dataset, we found in each case that normalized trial number was not a significant predictor (target first: p=0.33; lure first: p=0.55), while the interaction between stimulation site and stimulation condition was (target first: p=0.018; lure first: p=0.011).

Finally, we tested whether precise electrode targeting had an effect on the degree to which stimulation improved memory. Our electrodes were localized to three distinct regions containing fibers of the perforant path: the angular bundle (entorhinal white matter), gray matter of the entorhinal cortex (where perforant path fibers originate), and the nearby subiculum (which is perforated by the perforant path prior to its arrival in the hippocampus [*Zeineh et al., 2017*]). We hypothesized that, because stimulation to the perforant path can induce LTP, stimulating in the region where perforant path fibers are most densely concentrated might be the most effective. Specifically, we compared stimulation in the angular bundle, where perforant path fibers are most densely concentrated (*Yassa et al., 2010*; *Zeineh et al., 2017*), to stimulation in either the entorhinal or subicular gray matter, where these fibers are more spread out and fewer axons would be recruited by stimulation. Accordingly, we modeled the proportion of the remembered portraits as a function of two factors: stimulation region (whether the stimulating electrode was in the angular bundle or gray matter) and stimulation hemisphere (left versus right). We found significant main effects of each stimulation hemisphere (p=$1.49 \times 10^{-4}$, Wald $\chi^2$=14.39) and stimulation region (p=0.011, Wald $\chi^2$=6.53). The interaction between these factors was not significant (p=0.121, Wald $\chi^2$=2.41), indicating that the effects of stimulation on the right side and in the angular bundle each contributed to improved behavioral performance. Specifically, stimulation was most effective for improving memory specificity when the electrodes were positioned in the right angular bundle (EM = 0.16, 95% CI [0.09, 0.22]) (*Figure 4A*, *Figure 3—source data 1*; *Figure 4—source data 1*, *Source code 3*).

We next applied this expanded model to test the effect of stimulation on the other behavioral metrics defined above. As expected, DI—also a measure of memory specificity—was improved by right-sided stimulation (p=$2.79 \times 10^{-3}$, Wald $\chi^2$=8.94) and stimulation in the angular bundle (p=$2.53 \times 10^{-3}$, Wald $\chi^2$=9.12; interaction term: p=0.15, Wald $\chi^2$=2.07; right angular bundle EM = 0.15, 95% CI [0.08, 0.23]) (*Figure 4B*, *Figure 3—source data 1*; *Figure 4—source data 1*, *Source code 3*). A similar pattern was present in target acceptance, with stimulation in each the right side (p=0.034, Wald $\chi^2$=4.47) and the angular bundle (p=$2.52 \times 10^{-5}$, Wald $\chi^2$=17.75) significantly increasing target acceptance rates (right angular bundle EM = 0.074, 95% CI [0.0069. 0.1414]) (*Figure 4C*, *Figure 3—source data 1*; *Figure 4—source data 1*, *Source code 3*). A different pattern emerged when considering lure rejection rates. Here the interaction between stimulation region and hemisphere was significant (p=0.018, Wald $\chi^2$=5.62; right angular bundle EM = 0.080, 95% CI [0.025, 0.134]). The difference appears to be driven by stimulation in the right gray matter showing a trend toward impairing lure rejection (*Figure 4D*, *Figure 3—source data 1*; *Figure 4—source data 1*, *Source code 3*, *Source code 4*).

When considering lure rejection and target acceptance together, we found that stimulation on the right side increased the probability of correctly accepting previously viewed target images, regardless of the stimulation region. When right-sided stimulation was applied in the angular bundle, this also led to an increase in appropriately rejecting lure images. When applied in gray matter, however, stimulation led to an increase in incorrectly accepting lure images. Thus, it appears that right angular bundle stimulation improved memory specificity by simultaneously increasing the probability of accepting target images and rejecting lure images. Stimulation of the right gray matter, on the other hand—by increasing acceptance of both targets and lures—introduced a bias towards reporting positive memory, rather than increasing memory *per se*. Conversely, stimulation in the left entorhinal-adjacent gray matter decreased target acceptance rates and showed a moderate trend

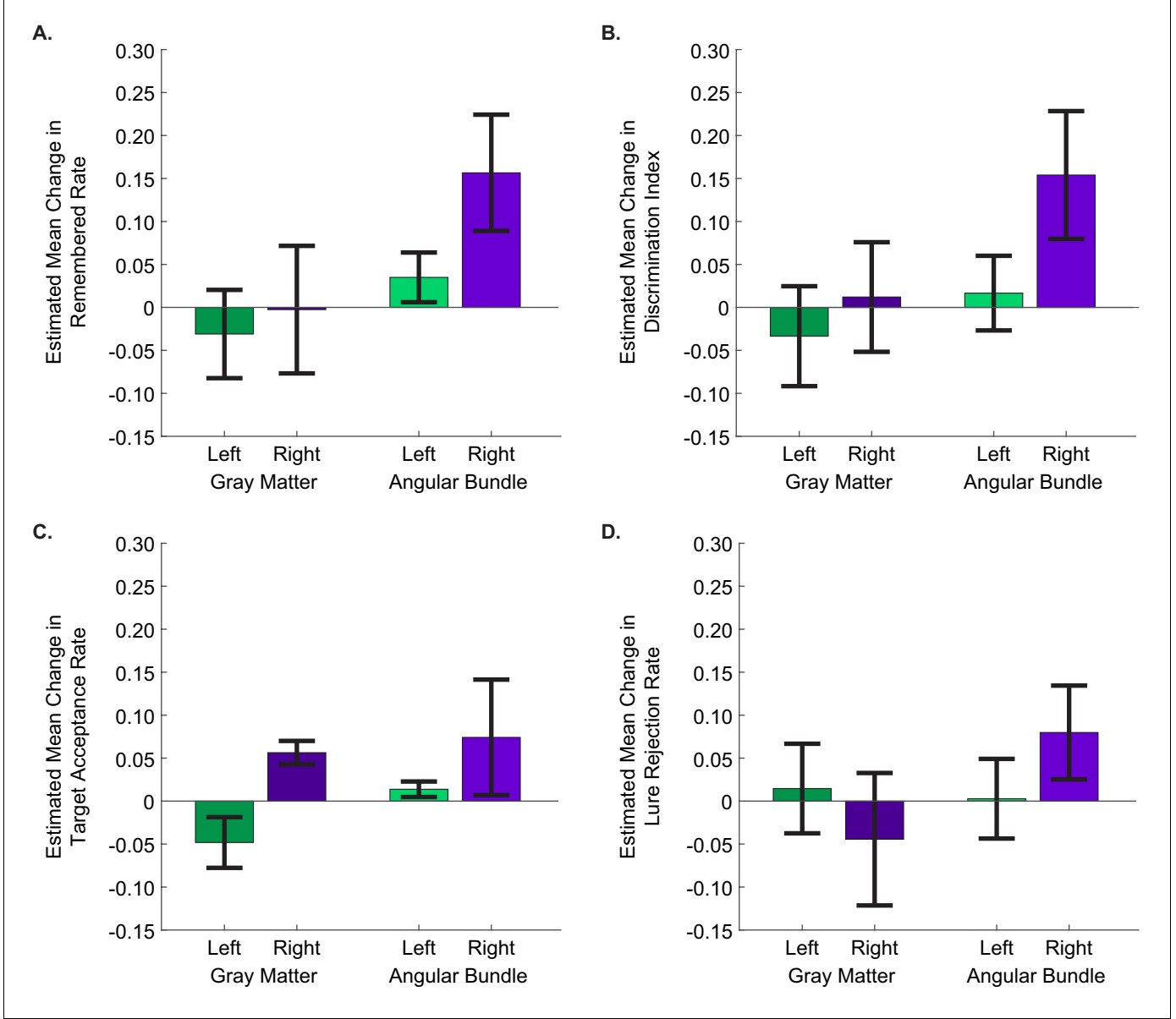

**Figure 4.** Stimulation of the right angular bundle improves memory specificity, whereas stimulation of gray matter introduces response bias. The estimated mean difference in stimulated compared to non-stimulated trials in (**A**) remembered rate, (**B**) discrimination index, (**C**) target acceptance rate, and (**D**) lure rejection rate, as computed by GEE models. In all panels, positive values indicate that performance was better for stimulated than non-stimulated trials. Error bars represent Wald 95% confidence intervals for the estimated means. The main effects of stimulation hemisphere and stimulation region were significant for all behavioral metrics except lure rejection rate, with stimulation in the right hemisphere and in the angular bundle improving performance for each (remembered rate: hemisphere: $p=1.49 \times 10^{-4}$, region: $p=0.011$; DI: hemisphere: $p=2.79 \times 10^{-3}$, region: $p=2.53 \times 10^{-3}$;target acceptance rate: hemisphere: $p=0.034$, region: $p=2.52 \times 10^{-5}$); In the case of lure rejection rate, the interaction between stimulation hemisphere and region was significant. ($p=0.018$). (Left gray matter: 11 sessions from three participants; right gray matter: 6 sessions from three participants; left angular bundle: 10 sessions from three participants; right angular bundle: 13 sessions from six participants.) (Data is in *Figure 3—source data 1*; SPSS model effects and definitions are in *Figure 4—source data 1* and *Source code 3*, respectively.).

DOI: https://doi.org/10.7554/eLife.29515.011

The following source data and figure supplement are available for figure 4:

**Source data 1.** Tests of Model Effects for session-level behavioral metrics.
DOI: https://doi.org/10.7554/eLife.29515.013

**Figure supplement 1.** Effects of stimulation in the entorhinal area, proper.
DOI: https://doi.org/10.7554/eLife.29515.012

toward decreasing lure acceptance rates, introducing a bias toward negative memory. Together these results suggest that in entorhinal/subicular gray matter, stimulation in the left and right hemispheres may introduce opposite biases.

Finally, it should be noted that, although we report uncorrected p-values throughout the results section, all reported statistically significant effects survive correction for multiple comparisons (see Materials and methods), further demonstrating the statistical robustness of our results. We also provide additional analysis in the supplement showing the main effects we report in *Figures 3* and *4* when data from the subiculum is excluded (*Figure 4—figure supplement 1*). The overall results are similar, but further studies will be necessary to examine potential differences between microstimulation of subicular and entorhinal cortices.

## Discussion

We found that microstimulation to the entorhinal area, targeted to maximally affect the entorhinal projections into the hippocampus, resulted in enhancement of memory specificity for images of novel people when the stimulating electrode was in the right hemisphere. This suggests a possible lateralization effect for memory of people, consistent with previous results: the right hippocampus shows increased metabolic activity during the learning of faces (*Haxby et al., 1996*), and single-unit activity in the right hippocampus encodes different facial expressions (*Fried et al., 1997*). While direct recordings of neural activity have shown evidence for lateralization, imaging studies have indicated that the encoding of pictorial material involves bilateral activation (*Kim, 2011*), and lesion studies noted a lack of asymmetry for encoding of non-verbal material (*Lee et al., 2002*). This discrepancy may indicate a difference between encoding of faces and visual encoding in general, as the latter studies were not specific to images of faces or people. On the other hand, our results indicate that that the left hippocampus is not uninvolved during facial encoding, as stimulation of the left entorhinal area had a behavioral effect, even if it was not to increase memory specificity. Previous findings of bilateral activation, therefore, may indicate only that the two hemispheres are both involved, not that they are performing the same function. Thus, these results warrant further studies into potential lateralization effects, both of function and of the effect of stimulation. It is important to note that our study was conducted in people with epilepsy, a condition which may disproportionately affect one side of the brain and could result in lateralized effects, such as impairment of cognitive skills lateralized to the affected hemisphere. Thus conclusions with respect to lateralization of function should be viewed with caution. However, we have repeated our analysis with data restricted to the sessions in which the stimulating electrode was positioned outside the clinically-determined seizure onset zone, and found qualitatively similar results.

Previous studies in animals have noted the importance of the frequency and mode of stimulation on behavioral outcomes. Specifically, theta-burst stimulation has been used to elicit LTP in a wide range of animal models from hippocampal slices (*Nguyen and Kandel, 1997*; *Staubli and Lynch, 1987*) to whole animals (*Larson et al., 1986*). In the current study, we present evidence supporting the efficacy of theta-burst microstimulation in protocols aimed at human memory improvement. Further research is warranted to determine whether theta-burst stimulation is more effective than other protocols, such as continuous stimulation. We also made no effort in this study to synchronize our stimulation protocols to endogenous theta rhythms in the hippocampus; we hypothesize that the memory effects we observed may be enhanced further by developing a closed-loop stimulation system, allowing the theta-burst pattern to be initiated on specific phases of theta.

Beyond the temporal dynamics of stimulation, we found that the spatial targeting of the stimulating electrode was also a critical factor in determining the effects of stimulation on behavior. Controlling this factor is more feasible with microstimulation than macrostimulation, as it is challenging to confine the spatial extent of delivered current when using macroelectrodes, due to their large contact surface area, wide inter-contact distance for bipolar stimulation, and high magnitude of stimulation current (*Figure 2—figure supplement 1*). In addition, high frequency stimulation delivered via macroelectrodes has been shown to inhibit nearby neuronal somata, while also providing excitation to axonal projections, indicating that small changes in electrode location could lead to substantially different results (*Herrington et al., 2016*). For example, stimulating in gray matter could disrupt neuronal computations, whereas electrodes that are confined to white matter may enhance excitatory drive to downstream regions (*Afraz et al., 2006*; *Arcot Desai et al., 2014*; *Fetsch et al.,*

*2014*; *Histed et al., 2009*). Future work examining the physiological effects of microstimulation in downstream regions, such as the hippocampus, would further enlighten the mechanisms of the differential effects when stimulation is applied in gray matter or the angular bundle.

Recently, studies on the effects of DBS—delivered via macroelectrodes in the entorhinal area—on memory performance have resulted in contradicting outcomes (*Jacobs et al., 2016*; *Suthana et al., 2012*). Stimulation delivered in these studies likely affected different neuronal assemblies, which—together with other methodological differences between them—may account for the variability in the reported results. Microstimulation is delivered via a single, small electrode, which eliminates the heterogeneity introduced in macrostimulation studies by variable inter-contact distances and implantation trajectories, and may make the final localization of the electrode easier to compare between research groups; this may increase the ease of replicability of studies using microstimulation, even in the face of other methodological differences.

Additional evidence for the importance of stimulation specificity arises from optogenetic studies in animals. In particular, a transgenic mouse model of early Alzheimer's disease shows that the more focally targeted an intervention is, the more specific of an effect can be elicited. This study demonstrated that direct optogenetic activation of specific hippocampal memory engram cells resulted in highly specific memory retrieval (*Roy et al., 2016*). Other optogenetic studies highlight the advantage of afferent stimulation to target downstream neuronal assemblies compared to direct stimulation of gray matter (*Gradinaru et al., 2009*; *Rajasethupathy et al., 2016*).

While optogenetics is the state-of-the-art method for targeting neural circuits with precise spatial and temporal resolution (*Rajasethupathy et al., 2016*), it is not likely to be feasible in humans in the near future. The microstimulation approach may, thus, be the method available in humans that most closely serves a similar purpose. Compared to macroelectrodes, microelectrodes have much smaller electrode contacts and affect a more localized region (*Arcot Desai et al., 2014*). The lower current levels typically delivered through microelectrodes have also been shown to excite, rather than inhibit, nearby neurons (*Afraz et al., 2006*; *Arcot Desai et al., 2014*; *Fetsch et al., 2014*), as well as those downstream of the nearby axonal projections (*Histed et al., 2009*). Moreover, high frequency microstimulation of the perforant path that induces LTP in hippocampus has been shown to cause widespread reorganization in hippocampal networks, as well as in cortical areas, such as medial frontal cortex and nucleus accumbens (*Canals et al., 2009*). These observations are compatible with the efficacy of microstimulation of the perforant path to cause overt behavioral changes in memory performance.

It has been suggested that stimulation in the entorhinal white matter, but not the adjacent gray matter, might lead to beneficial effects on memory (*Suthana et al., 2012*). Indeed, the results of the current study are consistent with this idea, such that applying microstimulation in the right angular bundle had a significant positive effect on memory for portraits, while applying microstimulation in the right entorhinal gray matter or subiculum did not improve memory, despite the presence of hippocampal afferents in these regions. This may be due to the fact that axons of the perforant path—a common locus for induction of LTP—are most densely concentrated in the angular bundle (*Yassa et al., 2010*; *Zeineh et al., 2017*) and hence a larger number would be recruited via stimulation. In parallel, stimulation in the gray matter may more directly impact or interfere with neuronal computations, which could drive the biases observed in our results due to gray matter stimulation. Although we hypothesize that LTP induced via the perforant path is likely responsible for the positive effects of right angular bundle stimulation, it must be borne in mind that the entorhinal region includes several additional fiber tracts—including the direct projections to CA1 via the alvear path, and hippocampal efferents—any of which may be affected by microstimulation, and current imaging techniques do not allow us to separate these various tracts.

Further, it is worth noting that the present study does not address the precise mechanisms by which microstimulation wields its behavioral influence. The distinct patterns observed as a result of stimulation in different hemispheres or different regions offer hints of a complex effect. Future studies investigating how neural signals change in response to microstimulation (and how these changes vary with the precise targeting of the stimulating electrode) will be critical for increasing our understanding, not only of the physiological signatures of microstimulation, but also the microcircuit dynamics underlying memory.

This study presents the first evidence that microstimulation has the potential to improve hippocampal-dependent memory in humans. In addition to the interesting implications for our

understanding of microcircuit dynamics of memory, this paves the way to developing new avenues that could one day be used for treating patients with chronic memory impairment. Chronic neural implants that use macrostimulation to treat conditions such as Parkinson's Disease and Epilepsy have become increasingly common (*Fisher and Velasco, 2014*; *Schuepbach et al., 2013*). As the use of neural implants moves toward treating cognitive disorders, one advantage of including micro-stimulation is the precise spatial targeting it affords, allowing for highly-controlled manipulation of neural circuits. Additionally, microstimulation requires lower levels of current to be delivered, thus allowing the implant to consume considerably less power. Future studies will be required to fully understand the stimulation protocols that provide the best cognitive enhancement for a wide variety of tasks. Large-scale datasets that evaluate the efficacy of macro- vs microstimulation, stimulation region and hemisphere, precise timing of stimulation relative to endogenous brain states, and more, will be required to establish the clinical relevance of stimulation for cognitive enhancement.

At present, there is only one relatively large-scale dataset published to date that attempts to report the effects of (macro) stimulation on memory (*Jacobs et al., 2016*). Even that study includes only five patients with entorhinal stimulation for one task (spatial memory) and seven for the other (verbal memory). Although that study reported an overall negative effect of stimulation on memory, it is important to note the many differences between it and the present study. In addition to the critical differences between micro- and macrostimulation described above, the tasks in the two studies were quite different. Here we investigated memory specificity in a person recognition task, while in the *Jacobs et al., 2016* the tasks required spatial navigation or memorization of a list of words, which may be differentially affected by stimulation. We also accounted for possible confounds of the effectiveness of stimulation, such as the precise location of the stimulating electrode and the hemisphere to receive stimulation. Without accounting for those factors, the positive effects of stimulation could be washed out.

In conclusion, our findings suggest that microstimulation, with its anatomical precision, physiologic-level currents, and action via axonal projections, holds promise for modification of memory circuits and thus for the treatment of memory impairments in people suffering from neurological disorders. More generally, this method may provide a tool for highly specific modulation of neuronal activity and human behavior (*Young and Deisseroth, 2017*).

## Materials and methods

### Study participants

The study subjects were thirteen patients (N = 13)* with pharmacoresistant epilepsy (*Figure 1—source data 1*) who met clinical criteria for depth-electrode placement in the entorhinal area for seizure localization and possible surgical cure by resection of the identified seizure focus. Electrodes fashioned with macro contacts along the shaft for intracranial EEG (Adtech Medical Instrument Corp., Racine, WI) and with micro wires at the distal end (California Fine Wire, Grover Beach, CA) were implanted stereotactically with the aid of CT angiography (CTA) and magnetic resonance imaging (MRI) (*Fried et al., 1993*; *1999*). Among the 13 subjects, 40 experimental sessions were conducted (n = 40), in which stimulation was applied in the right entorhinal area (N = 9, n = 19) or the left entorhinal area (N = 6, n = 21) (Two subjects performed sessions with right and sessions with left.)**. Language dominance testing was carried out when deemed clinically necessary. Language dominance was determined either through a standard Wada protocol or through fMRI language testing (*Połczyńska et al., 2015*). All research was carried out at the UCLA Medical Center and the UCLA Institutional Review Board approved the study protocol. All subjects provided written consent to participate in the study.

* Fourteen subjects participated in the study, however one subject was excluded due to severe psychological issues that arose during hospitalization and interfered with the patient's ability to complete any task.

** In one session (participant 13, left-sided stimulation), a small fraction of trials (17/52) received mistimed stimulation. These trials were excluded from analysis, but the session was included in the statistics reported in the main text. Additionally, we re-ran all statistics excluding this session entirely and the results remain nearly identical.

## Behavioral paradigm

Each experiment consisted of three phases: encoding, distraction, and retrieval. During the encoding phase, participants completed a behavioral task in which they viewed $31 \pm 12$ novel portraits of people (taken from *Versluis and Uyttenbroek, 2002*) on a computer screen (*Figure 1A*). People with epilepsy exhibit a wide spectrum of cognitive ability, due most likely to the effects of disease etiology as well as anti-epileptic drugs (AEDs). As such, we titrated the number of photographs shown for each subject with the guidance of their previous neuropsychological testing, as well as a short 4-portrait pre-test, using distinct stimuli, before the first experiment to ensure that the participants understood the task and could follow instructions. For our titrations, we targeted a 60–70% target acceptance rate to avoid ceiling or floor effects. During each encoding phase, for half of the images stimulation was applied for one second during the fixation period preceding image onset (*Figure 1A–B*); the subset of images to receive stimulation was selected randomly for each participant to avoid potential confounds based on stimulus complexity or order effects. After viewing the novel photographs, subjects performed a distractor task for 30 s in which single digits were presented serially, one per second, and subjects identified them as odd or even. This task is a non-mnemonic distractor task that has been shown to increase the demand for hippocampal involvement during the subsequent test phase (*Stark and Squire, 2001*). Finally, during the test phase, the same images from the encoding phase ('targets') were intermixed with an equal number of images of similar-looking portraits of different people ('lures') and presented one at a time (*Figure 1A,C*). In the test phase, participants were asked to identify whether each image was 'new' or 'old' by pressing a button. Participants also reported subjective confidence scores for each image, but these data are not presented. Target images were classified as '*remembered*' if the subject correctly accepted the target as seen before *and* correctly rejected the corresponding lure. Otherwise, images were classified as '*missed*'.

Behavioral performance outcomes were studied in four measurements: the proportion of pictures remembered, the proportion of targets accepted, the proportion of lures rejected, and discrimination index (DI), which is the proportion of correctly accepted targets minus the proportion of incorrectly accepted lure images (*Glosser et al., 1998*; *Suthana et al., 2015*).

## Microstimulation protocol

Monopolar constant-current microstimulation was applied using a BlackRock R96 Micro-Stimulator (BlackRock Microsystems, Salt Lake City, UT). The stimulation electrodes consisted of 100 µm diameter Formvar-insulated Platinum-Iridium (Pt/Ir) micro wires with the insulation removed from 1 mm around the tip, implanted into the right or left entorhinal area (*Fried et al., 1999*). Stimulation electrode impedance was measured immediately before testing to ensure it remained below 60 kΩ. Impedance values averaged $25.2 \pm 13.9$ kΩ. 150 µA cathodic-first, biphasic microstimulation was applied with a pulse width of 200 µs and an inter-pulse interval of 100 µs. Theta burst microstimulation pulse trains consisted of 4 such pulses at 100 Hz every 200 ms (*Figure 1B*), as this protocol has previously been demonstrated to be optimal for eliciting LTP in hippocampal slices from rat (*Larson et al., 1986*). This stimulation protocol resulted in a charge delivery of 30 nC per phase and a charge density of 9.32 µC/cm$^2$, well below the generally accepted upper safety limit of 100–150 µC/cm$^2$ for stimulation of neural tissue through Pt/Ir electrodes (*Merrill et al., 2005*; *Rose and Robblee, 1990*). Stimulation was applied for 1 s (5 pulse-trains composed of 4 pulses each) beginning at a time selected randomly from a uniform distribution between 2.2 and 2.7 s before picture presentation. All stimulation parameters were kept constant across patients, with the exception of precise electrode localization. Although two patients received stimulation on each side, the stimulating electrode was held constant throughout each experimental session. Electrode placements were confirmed post-surgically with co-registration of CT scans with preoperative MRI (See below; *Figure 2* and *Figure 2—figure supplement 1*). Sites of stimulation for each participant are summarized in *Figure 1—source data 1*.

## Brain imaging parameters

MRI data were acquired on a Siemens Magnetom Prisma 3 Tesla system housed in the Department of Radiology at UCLA. The whole brain MRI images were collected over 176 axial slices using a T1-weighted gradient echo sequence (TR 11 ms; TE 2.81 ms; flip angle 20 degrees; matrix size 256 ×

256 mm; FOV 256 mm; in-plane resolution 1 × 1 mm; slice thickness 1 mm; voxel size 1 mm isotropic). A high-resolution T2 weighted structural scan was also acquired for each subject (TR: 5300 ms; TE: 70 ms; flip angle: 178 degrees; matrix size: 500 × 500 mm; FOV: 200 mm; in-plane resolution: 0.4 × 0.4 mm; slice thickness: 3 mm, voxel size: 0.4 × 0.4 × 3 mm, 19 slices).

Spiral computed CT scans were performed on a 64-row multi-detector CT scanner. All scans had a pre-contrast series and single phase, post-contrast acquisition, synchronized using bolus tracking technique for arterial phase. Omnipaque 350 contrast media volume was set as 100 cc with an infusion rate of 3.0 cc/s.

## Electrode localization

A high-resolution post-operative computed tomography (CT) scan was co-registered to a pre-operative whole brain magnetic resonance imaging (MRI) and high-resolution MRI (*Figure 2*; *Figure 2— Figure supplement 1*) using BrainLab stereotactic localization software (www.brainlab.com; [*Gumprecht et al., 1999*; *Schlaier et al., 2004*]) and FSL FLIRT (FMRIB's Linear Registration Tool [*Jenkinson et al., 2002*; *Jenkinson and Smith, 2001*]). Medial temporal lobe regions (entorhinal, perirhinal, and parahippocampal cortices, and hippocampal subfields CA23DG [CA2, CA3, dentate gyrus], CA1, and subiculum) were delineated using the Automatic Segmentation of Hippocampal Subfields (ASHS [*Pluta et al., 2012*; *Yushkevich et al., 2010*]) software using boundaries determined from MRI visible landmarks that correlate with underlying cellular histology (*Amaral and Insausti, 1990*; *Duvernoy and Bourgouin, 1998*). In some cases, pixels that were clearly misplaced (e.g. single pixels from one region falling well within another or pixels that were entirely outside the MtL) were modified for aesthetics. In no case did this modification affect the outcome of the electrode localization procedure. In a single subject, whose electrode was farther anterior than ASHS can accommodate, subfields were delineated manually, with reference to brain atlases (*Amaral and Insausti, 1990*; *Duvernoy and Bourgouin, 1998*). White matter and cerebral spinal fluid areas were outlined using FSL FAST software (*Zhang et al., 2001*). Together, similar methods have been used previously to localize microelectrodes and investigate structural and functional dissociations within human medial temporal lobe subregions (*Ekstrom et al., 2008*; *Suthana et al., 2009*; *Zeineh et al., 2017*). A single 100 μm microelectrode at the distal tip, 3 mm from the most distal macro- contact, was used for microstimulation. Macro- and micro-electrode contacts were identified and outlined on the post-operative CT scan. To confirm white matter location of microelectrodes, the high-resolution MRI, with ASHS and FAST segmentation results, was overlaid with the co-registered electrode (*Figure 2*, *Figure 2—figure supplement 1*). For group-level electrode placements, individual localizations were mapped onto a standard MNI brain and visualized using the BrainNet Matlab toolbox (*Figure 2—figure supplement 2*) (*Xia et al., 2013*).

A neurologist reviewed clinical recordings in real-time while stimulation was applied to ensure absence of after-discharges. Participants did not report noticing any effects of stimulation, nor did they exhibit any seizures during pre-testing or task execution.

## Statistical analysis methods

Data were analyzed using SPSS (IBM Corporation, Armonk, NY) and custom scripts developed in Matlab (Mathworks, Natick, MA). All statistical models were implemented in SPSS using Generalized Estimating Equations (GEEs), which are a class of regression marginal models for investigating the relationships between clustered response data and outcome measures in a multivariate manner and a within-subject repeated measures design (Data and SPSS model definitions are in *Figure 3— source data 1–2*, *Source code 1–3*; Matlab code is in *Source code 4*). In our dataset, each subject performed the task a variable number of times (range: 1–7), due to circumstances that were beyond our control and unrelated to the experiment itself (most notably, the length of a patient stay in the hospital is determined by clinical criteria and can vary from several days to several weeks). Due to this variability, we sought a class of models that could accept different numbers of data points from each subject—without sacrificing the statistical power that could be derived from having multiple data points from most subjects—and GEEs were well-suited to the task (*Gardiner et al., 2009*; *Gueorguieva and Krystal, 2004*; *Hardin, 2005*; *Hubbard et al., 2010*; *Subramanian and O'Malley, 2010*).

Four session-level behavioral metrics were examined: the proportion of pictures remembered, DI, proportion of targets accepted, and proportion of lures rejected. For each of these, we computed the effect of stimulation on performance for each session by computing the metric for the subset of non-stimulated trials and subtracting that from the metric computed for the subset of stimulated trials. We implemented the models with participant identity as a within-subject variable, independent correlation matrices with robust covariance estimates, and identifying the metrics as linear scale responses, due to their approximately normal distributions. We included stimulation hemisphere—and later whether or not the stimulation electrode was in the angular bundle—as factors. We included model terms for these main effects, and an interaction term when applicable. Model statistics are reported in *Figure 4—source data 1*. Error bars in the figures (3A, 4A-D) denote Wald 95% confidence intervals for the estimated means.

For the trial-by-trial model, we selected a binomial logit model, as we had a two-level response variable (remembered vs missed), and used exchangeable correlation matrices. In addition to stimulation hemisphere, stimulation condition, and the interaction between these, we included whether the target was presented before the lure as a factor, and normalized trial number (trial number divided by the number of trials in the set) as a covariate. Model statistics are reported in *Figure 3—source data 3*. Error bars in *Figure 3C* denote Wald 95% confidence intervals for the estimated means.

Correction for multiple comparisons was performed using the Holm-Bonferroni method. To be as conservative as possible, we included all statistical tests that evaluated how stimulation's effect on memory depended on hemispheric and/or regional effects in a single correction test. This included the *a priori* tests from each figure panel in *Figures 3* and *4*. All tests that were originally reported as statistically significant at an alpha level of 0.05 were found to remain significant after this correction. Original, rather than corrected, p-values are reported in the text.

## Acknowledgements

We thank Tony Fields, Kirk Shattuck, Michael Jenkins, and Antonio Campos for technical assistance; Deena Pourshaban, Marianna Holliday, Samantha Briones, Nancy Guerrero, Güldamla Kalender, and Brooke Salaz for general assistance; the IDRE statistical consulting group at UCLA for providing insightful discussions regarding statistical analysis methods, and the participants for volunteering for this study.

## Additional information

### Funding

| Funder | Grant reference number | Author |
| --- | --- | --- |
| Schweizerischer Nationalfonds zur Förderung der Wissenschaftlichen Forschung | PBSKP3-124730 | Michael R H Hill |
| National Institute of Neurological Disorders and Stroke | NS084017 | Itzhak Fried |
| National Institute of Neurological Disorders and Stroke | NS058280 | Emily A Mankin Zahra M Aghajan |
| A.P. Giannini Foundation | Postdoctoral Fellowship | Emily A Mankin |
| Defense Advanced Research Projects Agency | N66001-14-2-4029 | Itzhak Fried |
| G. Harold and Leila Y. Mathers Charitable Foundation | 09212007 | Itzhak Fried |

The funders had no role in study design, data collection and interpretation, or the decision to submit the work for publication.

## Author contributions
Ali S Titiz, Formal analysis, Investigation, Writing—original draft, Writing—review and editing; Michael R H Hill, Conceptualization, Software, Funding acquisition, Validation, Investigation, Methodology, Critically reviewed manuscript; Emily A Mankin, Data curation, Software, Formal analysis, Funding acquisition, Validation, Investigation, Visualization, Writing—original draft, Writing—review and editing; Zahra M Aghajan, Data curation, Formal analysis, Validation, Writing—original draft, Writing—review and editing; Dawn Eliashiv, John Stern, Investigation, Methodology, Critically reviewed manuscript; Natalia Tchemodanov, Software, Investigation, Critically reviewed manuscript; Uri Maoz, Software, Formal analysis, Critically reviewed manuscript; Michelle E Tran, Data curation, Formal analysis, Investigation, Project administration, Critically reviewed manuscript; Peter Schuette, Formal analysis, Visualization, Critically reviewed manuscript; Eric Behnke, Visualization, Methodology, Critically reviewed manuscript; Nanthia A Suthana, Conceptualization, Supervision, Funding acquisition, Investigation, Methodology, Project administration, Writing—review and editing; Itzhak Fried, Conceptualization, Resources, Supervision, Funding acquisition, Investigation, Methodology, Writing—original draft, Project administration, Writing—review and editing

## Author ORCIDs
Emily A Mankin (iD) http://orcid.org/0000-0003-2163-913X
Zahra M Aghajan (iD) https://orcid.org/0000-0003-4245-3975
Itzhak Fried (iD) http://orcid.org/0000-0002-5962-2678

## Ethics
Human subjects: All research was carried out at the UCLA Medical Center and the UCLA Institutional Review Board approved the study protocol (IRB#10-000973). All subjects provided written consent to participate in the study.

## Decision letter and Author response
Decision letter https://doi.org/10.7554/eLife.29515.019
Author response https://doi.org/10.7554/eLife.29515.020

# Additional files

## Supplementary files
• Source code 1. Syntax to run model that generated *Figure 3A* in SPSS.
DOI: https://doi.org/10.7554/eLife.29515.014

• Source code 2. Syntax to run model that generated *Figure 3C* in SPSS.
DOI: https://doi.org/10.7554/eLife.29515.015

• Source code 3. Syntax to run models that generated *Figure 4*.
DOI: https://doi.org/10.7554/eLife.29515.016

• Source code 4. Matlab function to generate *Figures 3* and *4*.
DOI: https://doi.org/10.7554/eLife.29515.017

• Transparent reporting form
DOI: https://doi.org/10.7554/eLife.29515.018

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
