## [Decision Letter]

Thank you for submitting your article "Theta-burst microstimulation in the human entorhinal area improves memory specificity" for consideration by *eLife*. Your article has been reviewed by two peer reviewers, and the evaluation has been overseen by a Reviewing Editor and Sabine Kastner as the Senior Editor. The following individual involved in review of your submission has agreed to reveal her identity: Mary Pat McAndrews.

The reviewers have discussed the reviews with one another and the Reviewing Editor has drafted this decision to help you prepare a revised submission.

Summary:

The authors test whether theta-burst microstimulation of the entorhinal area can improve performance on a task that tests memory specificity, thought to be dependent on the hippocampus. They find that stimulation of the right entorhinal area improves the ability to recognize pictures of people and reject similar lures. They also find that stimulation of gray matter, as opposed to white matter, does not improve memory.

These findings could potentially make several important contributions to our understanding of how deep brain stimulation (DBS) using theta bursts can modulate episodic memory formation. In animal studies, such stimulation has been shown to induce long-term potentiation (LTP) thus there is a strong a priori reason to attempt to use it in human. The authors show that microstimulation can effectively improve memory encoding. This method could potentially have many advantages over the macroelectrode stimulation used in previous studies as microstimulation is much more spatially focused and uses currents on a more physiologic level. The authors also show that stimulation of gray matter did not improve memory. To some extent, this could be an explanation of previous contradicting results using macroelectrode stimulation (Suthana et al., 2012; Jacobs et al., 2016), which might either enhance or disrupt the memory pathway. Overall, the manuscript is well written and organized, however, there are several issues and questions that should be addressed so that a careful assessment of the contribution can be made.

Essential revisions:

1) Only two patients received entorhinal gray matter stimulation. Based on such a small number, it is not clear if these results will generalize. Additionally, these results could mean that the laterality effect is due to the fact that the majority of stimulation sites on the left side were in gray matter (also see below). Relatedly, in Figure 2, it looks like microelectrode for subject 1 is located in the entorhinal gray matter, not white matter, or at least it is in the area delineated as EC gray matter by ASHS. If this electrode is in fact in the gray matter, it would mean that only subject 4 received white matter stimulation on the left side. Looking at Figure 3, it looks like subject 4 actually showed a slight improvement on the task. Could this mean that the effect of laterality could be due to the stimulation site, not the hemisphere stimulated?

2) The 'positive' results (right EW with 5 of 6 showing the effect by 'eyeball' test on Figure 3) seems to be fairly compelling, whereas some of the 'negative' findings may be subject to serious power issues. This could be clarified by including number of sessions/subject in Figure 1—figure supplement1, but we only have subject-level information to go on (or aggregates in legend to Figure 4; whereas sessions contributed to effect estimations).

3) Subject 4 and Subject 10 have the opposite behavioral effect compared to others. What's the possible explanation for this discrepancy? Also, except these two subjects, 5 out of 8 subjects have been stimulated at the seizure-onset zone. Can such memory enhancement effect be generalized to healthy subjects?

4) Panel A in Figure 3 suggests no variance in left sites whereas panel B indicates there must be (as not all subjects showed 0 change). Figure 4 has such problem throughout all panels in that no variance (error bars) is depicted for the left gray matter group which actually has several participants whereas the left AB 'group' likely has 1 (based on Figure 2). These errors make statistics behind the plots questionable.

5) Since the authors postulated that theta-burst microstimulation would target afferent fibers and most directly affect downstream hippocampal subfields, it would be tremendously instructive and helpful to provide sample characterization of brain activity in the hippocampus after the stimulation. Without these recordings or data it is difficult to assess what the downstream impact on hippocampal dynamics might be.

6) In testing whether precise targeting of the stimulation had an effect on the degree to which stimulation improved memory, the authors did not sufficiently discuss varying the details of the stimulation protocol. It seems that every subject went through several sessions but it is not clear if the stimulation site was changed within the same session or across sessions. Is there an order effect of switching stimulation side?

---

## [Author Response]

Essential revisions:1) Only two patients received entorhinal gray matter stimulation. Based on such a small number, it is not clear if these results will generalize. Additionally, these results could mean that the laterality effect is due to the fact that the majority of stimulation sites on the left side were in gray matter (also see below).

Thank you for these comments. We have added data from left and right side in an additional subject. This subject completed a total of three sessions, with stimulation in the right entorhinal gray matter in two sessions and in the left angular bundle in one. These additions lend further support to the overall nature of our results while providing a more balanced sample. The nature of our patient pool means that we are not always able to select the side or precise site of stimulation, yet we trust that the addition of these data creates a more balanced dataset.

Relatedly, in Figure 2, it looks like microelectrode for subject 1 is located in the entorhinal gray matter, not white matter, or at least it is in the area delineated as EC gray matter by ASHS. If this electrode is in fact in the gray matter, it would mean that only subject 4 received white matter stimulation on the left side.

We thank the reviewers for noting our error. In fact, in this figure, we erroneously included the MRI image from the RIGHT entorhinal cortex, rather than the left. We have corrected this error and note that in the LEFT entorhinal cortex, the microelectrode is indeed in the white matter, so no statistics or analyses from the manuscript text were affected.

Looking at Figure 3, it looks like subject 4 actually showed a slight improvement on the task. Could this mean that the effect of laterality could be due to the stimulation site, not the hemisphere stimulated?

We now include a total of 3 subjects with stimulation of the left angular bundle (as noted above, subject 1 really did receive stimulation in the left angular bundle, and we have added another participant with left angular bundle stimulation), with 10 sessions between them. Using the updated model (see response to major point 4), we now find main effects of stimulation region and hemisphere. This indicates that stimulation site and hemisphere both contribute to the efficacy of stimulation, with the strongest effect observed when the right angular bundle is stimulated. We have updated the text to reflect these analyses.

2) The 'positive' results (right EW with 5 of 6 showing the effect by 'eyeball' test on Figure 3) seems to be fairly compelling, whereas some of the 'negative' findings may be subject to serious power issues. This could be clarified by including number of sessions/subject in Figure 1—figure supplement 1, but we only have subject-level information to go on (or aggregates in legend to Figure 4; whereas sessions contributed to effect estimations).

We have added this information to Figure 1—figure supplement 1.

3) Subject 4 and Subject 10 have the opposite behavioral effect compared to others. What's the possible explanation for this discrepancy? Also, except these two subjects, 5 out of 8 subjects have been stimulated at the seizure-onset zone. Can such memory enhancement effect be generalized to healthy subjects?

We appreciate the concern regarding whether the effects can be generalized to healthy subjects. While it is presently impossible to answer that directly, we did add an analysis to the manuscript showing that when using only the data in which the stimulating electrode is outside of the seizure onset zone (24 sessions from 9 subjects), the results of performing the left/right analysis are virtually identical to the original analysis. We have acknowledged this point and presented the results at the end of the discussion of Figure 3. Additionally, we confirmed that results are not overall changed for analyses presented in Figure 4.

4) Panel A in Figure 3 suggests no variance in left sites whereas panel B indicates there must be (as not all subjects showed 0 change). Figure 4 has such problem throughout all panels in that no variance (error bars) is depicted for the left gray matter group which actually has several participants whereas the left AB 'group' likely has 1 (based on Figure 2). These errors make statistics behind the plots questionable.

We thank the reviewers for bringing this to our attention. The model we used when we submitted this manuscript was chosen in consultation with statistics experts at UCLA (https://stats.idre.ucla.edu/). We chose, a priori, to use an exchangeable working correlation matrix. We acknowledge, though, that the error bars produced by this model in certain subsets of the data did not appear to reflect the true degree of variance present in the data. Thus, after further consultation, we decided to use independent working correlation matrices, instead. This was also – a priori – a plausible candidate for an appropriate working correlation structure. Indeed, this model produced similar expected means, with error bars that appear to better reflect the variance present in the data.

We note that although this change yields main effects for stimulation region and hemisphere, rather than statistical interaction, the overall conclusion—that microstimulation of the right angular bundle is the most effective for improving memory specificity—has not changed. Furthermore, we double checked that this difference is due to the model difference and not to the data added from the additional patient.

5) Since the authors postulated that theta-burst microstimulation would target afferent fibers and most directly affect downstream hippocampal subfields, it would be tremendously instructive and helpful to provide sample characterization of brain activity in the hippocampus after the stimulation. Without these recordings or data it is difficult to assess what the downstream impact on hippocampal dynamics might be.

We agree that without analysis of the downstream brain activity it is impossible to say with certainty what effects stimulation causes in the hippocampus. We have acknowledged this point in the discussion, highlighting the need for physiological analysis, particularly of the response to stimulation in downstream regions (Discussion section).

An in-depth analysis of the physiology is beyond the scope of the present manuscript. A thorough and careful analysis will take substantial additional time, and we believe that the behavioral findings presented currently are of sufficient importance to justify being published on their own. One reason we were motivated to publish these results in *eLife* is the “research advance” format offered here. We would like to publish the physiological analysis as a research advance to this article, which would allow the behavioral results to appear quickly, the physiology to be analyzed in depth, and proper authorship attribution to be ascribed to all involved in this paper and the next.

6) In testing whether precise targeting of the stimulation had an effect on the degree to which stimulation improved memory, the authors did not sufficiently discuss varying the details of the stimulation protocol. It seems that every subject went through several sessions but it is not clear if the stimulation site was changed within the same session or across sessions. Is there an order effect of switching stimulation side?

All stimulation parameters were kept constant for the duration of any experimental session. Apart from the location of the stimulating electrode, the stimulation protocol was kept constant across all sessions and patients. We have added explicit clarification of these points in the Microstimulation Protocol section of the Materials and methods section.

We include only two participants who received stimulation on each side, and the order was counter-balanced between them. Thus, we don’t believe that potential order effects between sets would be a confound for the effects we report.